# Direct Transfer of Mesoporous Silica Nanoparticles between Macrophages and Cancer Cells

**DOI:** 10.3390/cancers12102892

**Published:** 2020-10-09

**Authors:** Stefan Franco, Achraf Noureddine, Jimin Guo, Jane Keth, Michael L. Paffett, C. Jeffrey Brinker, Rita E. Serda

**Affiliations:** 1Internal Medicine, University of New Mexico Health Science Center, Albuquerque, NM 87131, USA; SAFranco@salud.unm.edu (S.F.); jiGuo@salud.unm.edu (J.G.); Kethj@unm.edu (J.K.); 2Department of Chemical and Biological Engineering, University of New Mexico, Albuquerque, NM 87131, USA; anoureddine@unm.edu (A.N.); jbrinker@unm.edu (C.J.B.); 3Fluorescence Microscopy Shared Resource, University of New Mexico Comprehensive Cancer Center, Albuquerque, NM 87131, USA; MPaffett@salud.unm.edu

**Keywords:** macrophage, mesoporous silica nanoparticle, doxorubicin, cancer, intercellular transport, tunneling nanotubes

## Abstract

**Simple Summary:**

A challenge for Nanomedicine is delivery to the site of action, most commonly the tumor microenvironment. When injected systemically, scavenger cells, such as macrophages, rapidly sequester nanoparticles. This study asks if macrophages can directly deliver scavenged nanoparticles to cancer cells via cellular connections known as tunneling nanotubes. Knowledge of the in vivo cellular fate of nanoparticles is critical for the design of optimized nanocarriers for clinical translation.

**Abstract:**

Macrophages line the walls of microvasculature, extending processes into the blood flow to capture foreign invaders, including nano-scale materials. Using mesoporous silica nanoparticles (MSNs) as a model nano-scale system, we show the interplay between macrophages and MSNs from initial uptake to intercellular trafficking to neighboring cells along microtubules. The nature of cytoplasmic bridges between cells and their role in the cell-to-cell transfer of nano-scale materials is examined, as is the ability of macrophages to function as carriers of nanomaterials to cancer cells. Both direct administration of nanoparticles and adoptive transfer of nanoparticle-loaded splenocytes in mice resulted in abundant localization of nanomaterials within macrophages 24 h post-injection, predominately in the liver. While heterotypic, trans-species nanomaterial transfer from murine macrophages to human HeLa cervical cancer cells or A549 lung cancer cells was robust, transfer to syngeneic 4T1 breast cancer cells was not detected in vitro or in vivo. Cellular connections and nanomaterial transfer in vivo were rich among immune cells, facilitating coordinated immune responses.

## 1. Introduction

As sentinels of the body, macrophages rapidly internalize pathogens and foreign materials, contributing to their rapid elimination from the blood [1]. While predominately thought of as a barrier to nanoparticle-mediated drug delivery, various groups have reported that macrophages can play a major role in nanomaterial delivery to sites of pathology. As an example, abundant migration of macrophages to sites of inflammation has led to the use of nanoparticles as magnetic resonance imaging agents, thus enabling monitoring of macrophage involvement in human diseases, including cancer, atherosclerosis, myocardial infraction, and diabetes [2]. In addition to inflammation, macrophages home to sites of neoangiogenesis [3,4], which is a hallmark of cancer [5], further supporting a role for macrophages in imaging and drug delivery for cancer [6]. For instance, lipid-surfactant-based nanoparticle-loaded macrophages have been reported to aid in the delivery of therapeutics across the blood–brain barrier [7], thus presenting new perspectives on mechanisms to target brain cancer.

While it was initially thought that cell-to-cell transfer of materials (e.g., prions and tau proteins) [8,9] reflects a pathological state, it is now believed that homotypic and heterotypic cell-to-cell connections are essential for diverse cellular processes, such as signal transduction, apoptosis, and immune responses [10]. The intercellular transfer of endogenous matter includes intact organelles, such as endosomes, lysosomes, and mitochondria; various chemical, electrical, or mechanical signals; and nanomaterials, enabling rapid communication and coordination of functions among neighboring and distal cells [11]. Homotypic and heterotypic transfer of material among immune cells facilitates coordinated defenses against pathogens, making cell-to-cell transfer highly beneficial in the stimulation of immune responses. For example, at immunological synapses, immune molecules such as MHC class I and class II proteins and costimulatory molecules undergo intercellular membrane transfer from antigen presenting cells (APC) to T cells [12,13]. Cells can communicate remotely via biovesicle transfer [14,15,16] but also by direct physical connections through open cytoplasmic channels that include tunneling nanotubes (TNTs) [17,18,19,20,21]. TNT formation can occur either *de novo* from filopodia-like protrusions, or during detachment of adjacent cells, with both processes being F-actin-dependent [22]. TNT-like structures have been observed bridging many immune cell types including B cells, natural killer cells, T cells, dendritic cells and macrophages [22,23].

Siliceous nanoparticles have been extensively used in cancer research as drug nanocarriers [24,25,26,27]. In 2011, Slowing et al. [28] reported asymmetric mesoporous silica nanoparticle (MSN)-transfer between endothelial cells and HeLa cells based on exocytosis by endothelial cells and reuptake by HeLa cells. In 2016, Rehberg et al. [29] demonstrated in vivo bidirectional movement through TNTs of another hard matter nanoparticle, carboxyl-modified quantum dots, between F4/80 positive macrophages in the skeletal muscle tissue of healthy mice. Our team [15] demonstrated the direct transfer of silicon microparticles between endothelial cells through TNTs. To date, there are no reports on the direct transfer of MSN between macrophages and cancer cells. Herein, we first demonstrate direct transfer of MSNs or their cargo between macrophages via TNTs containing tubulin, with localization of nanoparticle clusters existing in bulges within the TNTs termed gondolas. We then explore the ability of macrophages to transfer MSNs to human and murine cancer cells through cellular bridges as a potential means of drug delivery (Figure 1). In vivo biodistribution and co-localization of MSN with macrophages is explored using a syngeneic 4T1 mouse model of breast cancer following administration of free MSNs or adoptive transfer of MSN-loaded splenocytes.

## 2. Result and Discussion

### 2.1. Macrophages Internalize and Transport MSNs through Extensive Crosstalk

The RAW 264.7 macrophage-like (hereafter RAW) cell line, derived from the peritoneal fluid of a BALB/c mouse following transformation with murine Abelson leukemia virus [30], was used to study internalization and trafficking of MSN (200 nm diameter; 4 nm diameter pores, zeta potential = −34 mV) in macrophages and mechanisms of MSN and cargo transfer to surrounding cells, including cancer cells. 

To characterize MSN uptake/association with RAW macrophages, DyLight 488-conjugated MSN were added to cells, followed by imaging and quantitative flow cytometry analysis at 1, 3, or 24 h post addition. Analysis of percent positive macrophages by flow cytometry showed that MSN association with macrophages was both time and dose (10−100 µg/mL) dependent (Figure 2a). It is noteworthy that after only one hour, at least 50% of the RAW macrophages were associated with the negatively charged MSNs at the lowest dose (10 µg/mL), supporting highly efficient MSN association and uptake by macrophages. Relative intensity supported greater association per cell with increasing MSN concentrations.

Both filopodia (thin finger-like parallel bundles of filamentous F-actin [31]) outgrowth and TNT formation involve actin polymerization [32], with integrins and cadherins, often located in the tips of filopodia (i.e., sticky fingers), promoting adhesion of plasma membrane phospholipids to biological targets [33]. It has been demonstrated that filopodia in macrophages bind to soft biological pathogens and retract, pulling the pathogen towards the cell body [34]. Here we present scanning electron micrographs of surface topography showing macrophage association with MSN one hour following the addition of MSN to the cell culture media (Figure 2b–e). While Figure 2b shows an MSN aggregate (red) sitting on the cell membrane, supporting membrane-association before internalization, Figure 2c shows filopodia extending from a RAW macrophage to capture an MSN. Figure 2d,e similarly show MSN on the cell membrane, as well as on the membrane of TNTs connecting neighboring cells. Interestingly, filopodia originating from TNTs are also seen involved in MSN capture (Figure 2e).

Gerdes et al. [35] reported that TNT-like structures are associated with long filopodial protrusions in vivo, and it was suggested that these are precursors of TNT-like connections. More recently, Delage et al. [36] demonstrated that the filopodia-promoting CDC42/IRSp53/VASP network negatively regulates TNT formation and impairs intercellular vesicle transfer. Conversely, the actin regulator epidermal growth factor receptor (EGFR) pathway 8 (Eps8) has been shown to inhibit filopodia formation in neurons and positively regulate TNT formation. These latter data support the conclusion that despite structural similarities, filopodia and TNTs are regulated through different molecular mechanisms. Our data support a role for filopodia in the capture of MSN and a distinct role for TNTs in intercellular transport. 

### 2.2. Diverse TNT Assemblies and Gondolas

Extensive cellular connections among macrophages, both adherent and bridge type, are shown in the scanning electron micrographs in Figure 3a. Images were acquired using a 52 degree tilt and are presented at increasing magnification showing discrete areas of the cell-supporting silicon chip. Under these conditions, cellular bridges connecting the macrophages were typically 10−20 µm in length. The TNTs also existed in diverse geometries, with varying levels of connectivity. Figure 3b,c show secondary electron images of RAW cells following incubation with MSN for 5 h (to ensure > 80% MSN uptake, Figure 2a), at increasing magnification levels. TNTs were seen connecting cells via single or multiple points of attachment. Blue circles highlight three regions of TNT attachment, emphasizing the diversity in origins of the attachment. 

In addition to the existence of TNTs with varying lengths and diameters, TNTs expanded to accommodate large cargo, in regions referred to as gondolas (white arrows) [37]. Gondolas are shown in the highest magnification images, with MSN (either surface or beneath the cell membrane) shown in both grayscale and false-colored red in Figure 3b. These gondola-like blebs have been reported to move through cellular tethers mediating the transfer of contents between membrane-enclosed compartments [38]. The use of SEM surface topography to visualize MSN associated with cells is on occasion ambiguous since MSN seen on the cell surface could be: (1) sitting on the cell membrane, (2) in the process of internalization, or (3) exist just beneath the cell membrane (i.e., the dehydration processing causes cell shrinkage which could result in MSN protrusion of cell membranes). Alternative imaging with confocal microscopy supports the presence of MSN both within and on the cell surface. With respect to the mobility of MSN, the potential exists for movement both within and on the surface of TNTs. Bacteria have been reported to “surf” on the surface of TNTs based on the constitutive flow of the nanotube surface [39], providing a possible mechanism for external MSN movement. On the other hand, intracellular MSN are associated with microtubules, and based on their location in endosomes [40], transport is likely driven by microtubule motors, such as kinesins and dynein, or by actin-based myosin motors [41]. In summary, MSN can traffic both on the surface and within TNTs.

### 2.3. A Look Inside TNTs and Gondolas

Benefiting from cell surface cracking during sample processing (dehydration), the secondary electron micrograph in Figure 4a shows a TNT between two macrophages containing a split gondola. MSN within a gondola is false-colored to emphasize the abundance of MSN being transported across the TNT. Images were acquired using the upper secondary electron detector at a working distance of 4 mm using 10 kV energy. The two connected cells are seen in the upper left image in grayscale, while the two sequentially magnified images have the lower cell false-colored turquoise (Figure 4a). 

Using confocal microscopy, merged fluorescent, and DIC illumination images reveal cellular connections and material transfer between RAW cells co-cultured for 24 h following pretreatment with either 10 µg/mL DyLight 594-conjugated MSN (red) or CellTracker™ Violet (blue) (Figure 4b). The images confirm the presence of MSN or dye within TNTs or thicker substratum-tethered connections (white arrows). Other confocal images (Figure 4c) show an extensive microtubule network (anti-tubulin Alexa Fluor 488, green; nuclei in blue) within and between macrophages. MSN (DyLight 647-conjugated, red, indicated by white arrows in center image) can be seen within a cellular bridge in 2D and 3D images (arrows, Figure 4c). The 3D image is sectioned to highlight the location of MSN within the microtubule (green)-filled TNT). Rotation of the 3D image and removal of the merged bright-field enabled us to visualize the location of MSN relative to tubulin within the bridge connecting the two cells. Three consecutive image slices show microtubules within the TNT extending into the microtubule networks of both cells. 

### 2.4. The Impact of Environmental Triggers on Intercellular Transfer of MSN through TNT

Various stressors placed on cells have previously been implicated as one mechanism to amplify the intercellular transfer of material between cells. We previously reported that serum starvation increases the rate of silicon microparticle transfer between endothelial cells and that cytochalasin B inhibits the formation of TNT in macrophages [15]. Zhu. et al. [42] reported that hydrogen peroxide increases TNT formation in astrocytes with the co-localization of myosin-Va with F-actin supporting activation of active transport mechanisms. Wang et al. [43] demonstrated that ultraviolet (UV) light induced the formation of microtubule-containing TNTs between stressed and non-stressed PC12 cells, with the preferential transfer of mitochondria from healthy to stressed cells. While they report that healthy PC12 cells were able to rescue UV stressed cells, another study [44] showed the transfer of undamaged organelles, pathogens, and cytotoxic agents from stressed cells. The commonality in these studies is that TNT mediated intercellular transfer results in a reduction in stress fostering cell rescue.

Herein, we compared MSN transfer from unstained to stained acceptor cells in the presence of various stressors using flow cytometry. The impact of inflammation (1 µg/mL lipopolysaccharide, LPS), hyperthermia (42 °C for the final 30 min of incubation), or serum starvation on MSN transfer between macrophages was studied through co-culture of RAW cells previously stained with either CellTracker™ Violet or pre-incubated for 5 h with 10 µg/mL DyLight 594-conjugated MSN, with extensive washing of the latter before co-culture at 37 °C for 24 h. The rate of transfer was set to 1 for standard culture conditions (10% serum, no LPS, 37°C), with relative transfer under the various conditions shown in Figure 4d. While LPS (1 µg/mL) did not stimulate any observable changes in transfer, hyperthermia reduced the rate of transfer, and serum starvation increased MSN transfer nearly 2-fold. The combination of LPS and interferon-gamma also did not alter the rate of MSN exchange (data not shown). We did not evaluate if this reflects an increase in the MSN transfer rate or number of TNTs. Zhu et al. [45] reported an increase in TNTs between Schwann cells during serum starvation, whereas Kumar et al. [46] reported that no differences exist in the frequency of TNT formation for A549 lung cancer cells grown in low serum or complete media. 

### 2.5. Trans-Species Transfer of MSN from Murine Macrophages to Human Cancer Cells

The formation of multicellular tumor networks containing cancer cells connected by membranous structures has been correlated with prognostic features of malignant brain tumors [47]. Wiskott–Aldrich syndrome protein (WASP), a paracrine factor important in interactions between macrophages and breast carcinoma cells, has been shown to have a role in TNT formation, tumor cell invasion, and metastasis [48]. Here, we investigated trans-species connections and MSN transfer between murine RAW macrophages and human HeLa cervical cancer cells. The existence of homotypic TNTs among HeLa cells is shown in the false-colored SEM image in Figure 5a. A heterotypic connection between a HeLa cell (red) and macrophage (white) is shown at two magnifications in SEM images in Figure 5b. False-coloring of images is subjective, and discrimination of cell type was based on findings that macrophages were relatively small and round compared to cancer cells. In addition, intercellular connections differ between cell types, as described in the next paragraph. Grayscale images are included in addition to false-colored images for mixed culture images. Elaborate points of connection existed on the surface of the cancer cell, with the cancer cell appearing to form a pedestal, several micrometers in diameter, at the site of contact with the macrophage.

The rate of homotypic and heterotypic MSN transfer between murine macrophages and either naive macrophages or human cancer cells was studied. Discrete populations of RAW cells were preloaded with either PromoFluor (PF) −633-labeled MSN, or DyLight 488-labeled MSN, with the latter also used to preload HeLa or A549 cells. After extensive washing to remove free MSN, the populations were mixed, and cells were co-cultured either in standard media or in media supplemented with IL−12 (0.1 µg/mL) for 24 h (Figure 5c, upper graph). Similar to the negligible impact of LPS addition (Figure 4d), the pro-inflammatory cytokine IL−12 did not alter RAW to RAW MSN transfer rates. Surprisingly, based on flow cytometry analysis of double-positive cells, MSN transfer between RAW and HeLa cells, or between RAW and A549 cells, was significantly greater (*p* < 0.01, unpaired *t*-test) than that observed between RAW cells.

To further characterize the nature of interactions between RAW and HeLa cells, secondary electron micrographs were captured after 24 h of co-culture (Figure 5d). Based on the size and surface topography, macrophages and HeLa cells were pseudo-colored white and red, respectively, in the lower row of images. While the diameter of TNTs between macrophages was typically less than 200 nm and non-adherent (white arrow), interspecies connections between macrophages and HeLa cells were generally thicker (red arrow; 1.4 µm in diameter), and included both bridged and adherent connections. The sequential magnification of cells was used to highlight the abundance of interconnectivity between the different cell and species types. Similar to the cells presented in Figure 5b, Figure 5d shows an elaborate point of connection between the human cancer cell and the murine macrophage. The elaborate membrane structure existed on the thick adherent connection and has, therefore, been colored as originating from the cancer cell. The biological significance of these extravagant connections involved in heterotypic TNT formation between macrophages and cancer cells is unknown”.

### 2.6. Transfer of Carrier-Delivered Chemotherapeutics between Macrophages

As stated previously, rapid internalization of nanoparticles by macrophages and trafficking to sites of inflammation has led to reports of macrophages functioning as cell-based drug carriers [49,50,51]. Zhang et al. reported that silica-based nano capsules loaded with doxorubicin can be loaded into macrophages with minimal drug release in the first hours after uptake and minimal effect on in vivo cell migration to tumors [52]. In further support of targeting macrophages as drug and nanocarriers, here we studied the intercellular transfer of doxorubicin between macrophages using a second macrophage-like cell line, murine J774A.1. J774A.1 cells were pretreated with doxorubicin-loaded mesoporous silicon microparticles (1000 × 400 nm, discoidal) for three hours, followed by the acquisition of live cell fluorescent and DIC images every 10 min for 12 h. Select DIC and fluorescent merged images are presented in Figure 5e. The time-series images show two red fluorescent doxorubicin-loaded macrophages directly bound to each other 2 h after imaging was initiated. After an additional 3 h, the two cells have separated, but remain in contact via a doxorubicin-containing TNT. Widespread fluorescence within the TNT (non-punctate), and previous work showing a characteristic doxorubicin release lifetime of 2 h from mesoporous silicon microparticles [53], support TNT-mediated transfer of free drug between macrophages. Desir et al. [54] reported that doxorubicin stimulates TNT formation among pancreatic cancer cells, facilitating the intercellular redistribution of the drug. Such exchanges of chemotherapy could promote cancer progression by enabling drug efflux. Drug efflux/dilution could similarly promote cell survival among macrophages, as well as progress information among immune cells on the nature of invaders.

### 2.7. Lack of Murine Intraspecies MSN Heterotypic Transfer from Macrophages to Breast Cancer Cells

The majority of both murine macrophages (93%) and 4T1 breast cancer cells (83%) were able to internalize fluorescent MSN. However, uptake was greater for macrophages (1.8 fold) based on relative fluorescence detected using flow cytometry (Figure 6a). Unlike the elaborate cytoplasmic connections between RAW and HeLa cells, or among RAW cells, confocal and electron micrographs show RAW cells bound directly to the surface of syngeneic 4T1 breast cancer cells, in multiplicity (Figure 6b,c). Following 24 h co-culture of CellTracker Violet labeled 4T1 cells and DyLight 594-labeled MSN (red) loaded-macrophages, MSN remained associated with macrophages (Figure 6d). 3D merged confocal and DIC images show 4T1 cells (blue) surrounded by macrophages (left). To the right, macrophages with DAPI labeled nuclei (blue) are seen surrounded by a ring of fluorescent MSN (red), while the larger CellTracker Violet labeled 4T1 cells lacked detectable MSN and TNT connections, supporting a lack of MSN transfer between RAW macrophages and syngeneic 4T1 cancer cells. While the data did not support MSN transfer among 4T1 murine breast cancer cells or from RAW macrophages to syngeneic 4T1 cancer cells, Guo et al. [51] reported that doxorubicin-loaded M1 RAW macrophages transfer drug cargoes into human SKOV8 and mice ID8 ovarian cancer cells. However, while RAW macrophages and 4T1 cells are derived from BALB/c mice, ID8 cells are derived from C57BL/6 mice. Therefore, the Guo et al. data support TNT formation and drug exchange between allogeneic murine macrophages and cancer cells. Fu et al. [55] reported that doxorubicin-loaded RAW macrophages displayed anti-cancer efficacy in a 4T1 mouse model. However, the mechanism was described as the release of the drug from macrophages into the surrounding microenvironment. In summary, the existence and biological relevance of TNT formation and drug/MSN exchange between syngeneic macrophages and cancer cells require further study.

### 2.8. In Vivo MSN Biodistribution and Trafficking in TNTs

#### 2.8.1. Biodistribution of Free MSN and Splenocytes-Loaded MSN

Here macrophages (syngeneic splenocytes) were used as trojan horses to evaluate their ability to deliver MSN to tumor sites in vivo compared to the trafficking of free MSN. The IVIS Spectrum imaging system was used to track DyLight 633 or 800 labeled MSN or carrier macrophages in vivo and subsequently to image MSN-organ association. Incubation of harvested BALB/c splenocytes with fluorescent MSN resulted in a time-dependent increase in internalization from 1−24 h (Figure 7a). Images taken 24 h following intravenous administration of MSN, or MSN-loaded splenocytes, to female BALB/c mice bearing orthotopic 4T1 tumors 24 h following injection of 150 µg free MSN (“free”, Figure 7) or adoptive transfer of MSN-loaded splenocytes (“cells”, Figure 7) resulted in MSN predominately localized in the filtering organs (liver and spleen) and tumor (Figure 7b). It has been reported that approximately 95% of nanoparticles administered intravenously to mice accumulate in filtering organs [56]. Here, quantitation of the fluorescent signal in extracted organs confirmed that the majority of MSN were located in the liver irrespective of injection mode (free vs. splenocyte-loaded MSN) (Figure 7c). The amount of MSN in the tumor relative to the liver was similar for free verses cell-carried (adoptive-transferred) MSN. This is consistent with rapid intravenous uptake of injected free MSN by endogenous macrophages and cell-based trafficking of both MSN populations. In 2018, W. Chan and colleagues reported that within tumors, the majority of nanoparticles were either in the extracellular matrix or internalized by perivascular tumor-associated macrophages [57]. 

#### 2.8.2. MSN Association with Hepatic Macrophages and In Vivo Homotypic MSN Transfer via TNTs

Based on the aforementioned findings, we then compared in vivo macrophage association with MSN based on delivery mode [free or cell-carried (i.e., adoptive transfer)] using immunofluorescence (24 h post intravenous administration). The scavenger receptor CD204 signifies an M2 phenotype [58], while F4/80^+^ is a commonly used marker for monocytes and tissue macrophages [59,60]. OCT frozen tissue sections were stained with DAPI (nuclei, blue), and either all cells were visualized based on autofluorescence (green) or macrophages were labeled using Alexa Fluor 488 F4/80 or CD204 antibodies. In hepatic tissue, adoptive transferred MSN (red) were found within cellular bridges connecting adjacent cells (Figure 8a; white arrows). In the spleen, adoptive transferred MSN were found arranged in rings (Figure 8a, bottom image), supporting localization in marginal zone dendritic cells or macrophages [60,61]. These myeloid cells are known for the clearance of microbes and antigens, suggesting that either endogenous phagocytes internalized the injected MSN-loaded splenocytes or the injected myeloid cells migrated to the marginal zone. 

An abundance of adoptive transferred MSN (red) were found in vivo associated with cells positive for either F4/80 or CD204 (green) (Figure 8b,c). Antibody-labeled cellular bridges were visible between both F4/80 and CD204 hepatic macrophages (yellow arrows). Administered free MSN were also associated with macrophages. Figure 8d shows single and merged fluorescent micrographs of MSN (red) located within hepatic CD204 positive macrophages (green). The magnified region (lower left) shows a CD204 antibody-labeled (green) cellular connection containing MSN (Figure 8d; white arrow). No MSN were visible in the examined tumor sections. Independent of delivery route, MSN in the liver were associated with macrophages indicating that this is the main population of cells engulfing MSN, making in vivo cellular interactions and material transfer essential for the MSN delivery of cargo.

To further examine in vivo cellular connections among immune cells, scanning electron micrographs were acquired of murine lymphatic tissue. Tissue sections from the inguinal (draining) lymph node of a BALB/c mouse are presented in Figure 9. Elaborate cellular connections can be seen between both adjacent (middle image; T cells) and distant (right image, APC) cells. False coloring was used to highlight actively communicating cells.

## 3. Materials and Methods

### 3.1. Cell Culture

HeLa cervical cancer cells, A549 lung cancer cells, RAW 264.7 and J774A.1 macrophage-like cells were cultured at 37 °C in 5% CO_2_ in the vendor recommended media. Media and fetal bovine serum were purchased from Invitrogen Corporation (Carlsbad, CA, USA) and ThermoFisher Scientific (Grand Island, NY, USA), respectively. 

### 3.2. MSN Fabrication

Mesoporous silica nanoparticles (MSN; 200 nm particle size with 4 nm pore) were purchased from Sigma–Aldrich or fabricated in house, as previously described [62]. NHS-ester-activated DyLight fluorophores (ThermoFisher) were conjugated to propylamine functionalized MSN using 50−65 µg DyLight for every 2.5 mg MSN in ethanol at 10 mg/mL, with a one-hour incubation at room temperature with rotation. 

### 3.3. Confocal Microscopy Imaging of TNT Transport of Nanoparticles 

RAW 264.7 macrophages were seeded onto glass coverslips in 6-well plates at a density of 5 × 10^5^ cells per well. After 24 h incubation, fluorescent MSN were added in fresh complete media at 10 µg/mL. After the indicated incubation times, cells were washed with PBS, fixed with 4% paraformaldehyde in PBS for 15 min with prewarmed solutions followed by overnight refrigeration, rinsed twice with PBS, and permeabilized with 0.1% Triton-X in PBS for 15 min. Cells were then blocked with 1% BSA for 20 min and then labeled with 5 units/0.5 mL Alexa Fluor 647 phalloidin and/or 10 µg/mL mouse anti-α-tubulin antibody-Alexa Fluor 488 in 1% BSA for 1 h. After rinsing with PBS, slides were mounted using Prolong Gold with DAPI. Confocal images were acquired with a 63×/1.4NA oil objective in sequential scanning mode using a Leica TCS SP8 confocal microscope.

For co-cultures, HeLa, A549, or 4T1 cells were seeded onto coverslips in 6-well plates at a density of 0.5 × 10^5^ cells/mL. After 24 h, cells were either incubated with 10 µg/mL fluorescent NPs or labeled with Vybrant DiO for 8 min at 37 °C in serum-free media at 5 µl/mL. RAW 264.7 cells, pre-incubated with 10 µg/mL fluorescent NPs for 2−3 h, were either stained with CellTracker Violet, as described by the manufacturer, or washed and added directly to adherent HeLa or RAW 264.7 cells. Cells were either incubated in standard media; in media supplemented with LPS (1 µg/mL) or IL−12 (0.1 µg/mL), or placed in a 42 °C water-bath for 30 min of the incubation time. After the indicated incubation time, cells were fixed and permeabilized as previously described. Cells were then blocked with 1% BSA for 20 min and then labeled rabbit anti-α-tubulin antibody-Alexa Fluor 647 or anti-mouse CD14 FITC at 2.5 µg/mL in 1% BSA for 1 h. After washing with PBS, slides were mounted using Prolong Gold with DAPI or Vectashield Antifade Mounting Medium with DAPI. Images were taken using a Leica SP8 confocal microscope equipped with a 60× oil immersion lens.

### 3.4. Live-Cell Microscopy

Oxidized silicon microparticles (discoidal shaped, 1000 × 400 nm) were loaded by immersion of the dry microparticles in a concentrated solution of doxorubicin in water for 30 min followed by washing in water. Murine J774A.1 macrophages were incubated with doxorubicin-loaded porous silicon microparticles (40 per cell) for 2 h, then washed, and co-cultured with naïve J774A.1 cells (complete media; 37 °C). Bright-field and fluorescent images were captured every 15 min over 12 h using the Olympus IX81 automated inverted microscope equipped with an SRV CCD Digital Camera and CO_2_/humidity/temperature control incubator (Olympus, Melville, NY, USA).

### 3.5. Scanning Electron Microscopy (SEM) Imaging of Tissue and Cells 

Tissue was collected in 2.5% glutaraldehyde, followed by embedding in 3% agarose. Tissue sections of 100 mm thickness were cut using a Krumdieck MD−4000 Tissue Slicer (Alabama Research & Development, Munford, AL, USA). Sections were rinsed with cacodylate buffer, immersed in cacodylate-buffered 2% tannic acid for 24 h, and washed twice with 0.2 M sodium cacodylate and then incubated in cacodylate-buffered 2% osmium tetroxide for 2 h at 4 °C, washed again in 0.2 M sodium cacodylate, and dehydrated in increasing concentrations of ethanol, followed by infiltration with 100% t-butanol. Samples were dried in a desiccator and then mounted on SEM sample stubs using carbon adhesive tape.

RAW 264.7 macrophages, 4T1, or HeLa cancer cells were seeded in 24-well plates containing 5 × 7 mm silicon chip specimen supports (Ted Pella, Inc., Redding, CA, USA) at 1 × 10^5^ cells per well. Cells were then incubated with 10 µg/mL 200 nm MSN for 1, 3, or 24 h and then processed for SEM imaging, as previously described [63]. Alternatively, HeLa or 4T1 cancer cells were seeded onto silicon chips, and the next day RAW cells, preloaded with NPs, were added, and cells were incubated for an additional 24 h. SEM images were acquired under a high vacuum, at 1−30 kV, using a Hitachi SU8230 Scanning Electron Microscope (Hitachi High Technologies, Clarksburg, MD, USA) or an FEI Quanta 3D FEG (FEI, Hillsboro, OR, USA). Low voltage imaging was performed without sputter-coating using the Hitachi SU8230, while high voltage imaging was performed on samples sputter-coated with approximately 5 nm gold or gold-palladium. Site-specific milling with the FEI Quanta 3D FEG was performed using a large rough (30 pA) cut to eliminate one cell and fine cut (10 pA) to open the TNT at the gondola. Some images have been pseudo-colored using Adobe Photoshop (Adobe Systems Incorporated, San Jose, CA, USA) and gamma levels adjusted to enhance image contrast and brightness.

### 3.6. Flow Cytometry Analysis of Particle Association with Cells and Transfer between Cells

RAW macrophages were seeded in 6-well plates at 2.5−5 × 10^5^ cells per well and allowed to adhere overnight. Cells were then incubated with 10−100 µg/mL fluorescent silica NPs for 1, 3, or 24 h for cell uptake studies or 2−5 h for cell transfer studies. For cell transfer experiments, unique cell cultures were stimulated as indicated and incubated with DyLight 488 or 594 or DiO, as described by the manufacturer for 3−5 h. The loaded cells were then extensively rinsed with sterile PBS to remove any free particles, scrapped, and plated with cells pre-incubated with NPs with a distinct label or CellTracker™ Violet. Cells were co-cultured as indicated and analyzed using either a BD™ LSRII, Fortessa, or FACSCalibur flow cytometer using FACSDiva or CellQuest™ software (BD Biosciences, San Jose, CA, USA). 

### 3.7. Biodistribution and Tissue Confocal Microscopy

BALB/c mice with approximately 0.25−0.5 cm 4T1 breast tumors, originated in the mammary fat pad using 1 × 10^5^ cells in PBS, were administered fluorescent NPs or splenocytes preloaded with NPs intravenously via retro-orbital injection (150 µg/mouse). Mice were anesthetized 24 h post-injection and sacrificed by cervical dislocation. Tissues were frozen in O.C.T. on dry ice, stored at −80 °C, and sections stained fixed in 100% ice-cold acetone when ready for use. Following blocking in 1% FCS in PBS, tissues were stained using rat anti-mouse F4/80 or CD204 Alexa Fluor 488 antibodies at a dilution of 1:50 in blocking buffer for 1 h. Tissues were then washed with PBS and mounted in Prolong Gold with DAPI. Images were acquired using a Leica SP8 confocal microscope equipped with a 60× oil immersion lens.

### 3.8. Statistical Analysis

Experimental groups were compared using nonparametric, unpaired Student’s *t*-tests.

## 4. Discussion

This study investigated the capability of macrophages to transfer MSNs to either homotypic or heterotypic cells through TNTs and further exploit this feature to provide visual evidence of TNT-mediated MSN transfer from carrier macrophages to cancer cells as a means of drug delivery. Herein, we have demonstrated the involvement of cellular bridges in MSN transfer between macrophages both in vitro using RAW macrophages and in vivo following administration of free or cell-carried MSN. Our data also support a role for the external conditions of the microenvironment (temperature and presence of serum) in influencing MSN transfer between cells. 

Our findings further support high rates of in vitro MSN heterotypic transfer between murine macrophages and human HeLa and A549 cancer cells, with abundant trans-species bridges connecting the cells. Interestingly, no evidence of MSN transfer from RAW macrophages to syngeneic 4T1 breast adenocarcinoma cells was observed. Rather than forming heterotypic TNTs, macrophages and 4T1 cancer cells interacted through side-by-side adherence. While others have reported TNT formation between macrophages and cancer cells, the studies were restricted to allogeneic or trans-species cells, wherein cells are immunologically incompatible. 

## 5. Conclusions

While homotypic TNT formation among macrophages is likely to represent normal gatekeeper functions, heterotypic membrane connections with allogeneic or trans-species cancer cells may be triggered by an allorecognition immune response. The defense system has an inherent ability to recognize and reject cells of genetically disparate origin. While T cells play a central role in the mammalian allogeneic response, innate immune cells, including myeloid cells, can mount an attack on cells missing self-MHC (major histocompatibility complex) or based on recognition of allodeterminants not linked to MHC [64]. In this study, it is interesting that the thick cellular connections between human HeLa cells and murine macrophages appear to originate from the cancer cells (based on contiguity of the elaborate points of connection with the cancer cell membrane). Perhaps rather than a myeloid response, TNT formation and material transfer can be initiated by the cancer cell in response to signals released by activated immune cells. Further research into environmental cues present during allorecognition by myeloid cells could potentially be used to stimulate TNT formation and nanomaterial transfer in syngeneic/autologous tumor environments.

## Figures and Tables

**Figure 1 cancers-12-02892-f001:**
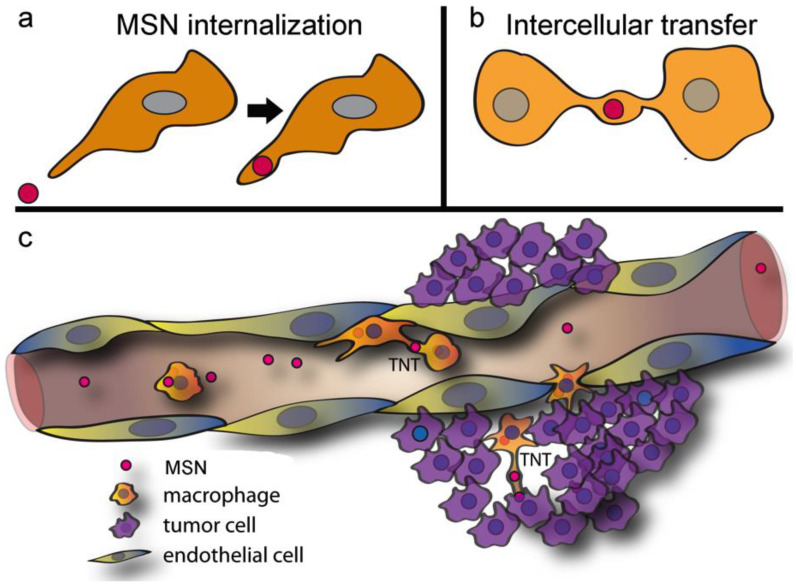
Proposed in vivo trafficking of mesoporous silica nanoparticle (MSN) to the tumor microenvironment. (**a**) MSN administered intravenously was rapidly internalized by systemic macrophages. (**b**) Macrophages are highly dynamic and interactive, with intercellular connections, known as tunneling nanotubes (TNT), enabling direct cell-to-cell transfer of MSN to neighboring or distant cells. (**c**) Proposed movement of MSN to the tumor microenvironment.

**Figure 2 cancers-12-02892-f002:**
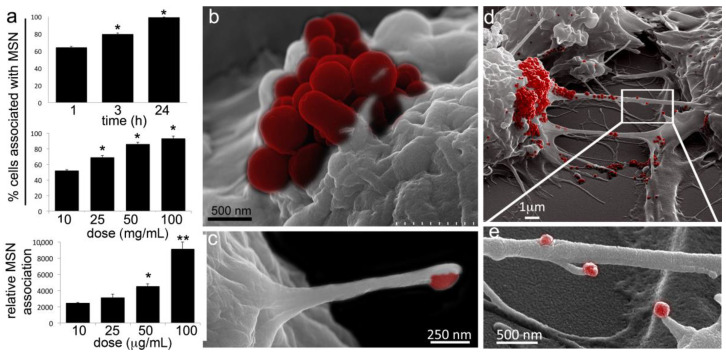
Murine macrophages robustly internalize MSN. (**a**) Flow cytometry analysis of cell association with fluorescent MSNs following incubation with 10 µg/mL DyLight 488-conjugated-MSN for 1, 3, or 24 h at 37 °C (top graph). Percent of cells positive for fluorescent MSN association (middle graph) or mean fluorescent intensity (MFI; bottom graph) of cells 1 h after the addition of 10−100 µg/mL MSN. (**b**–**e**) Pseudo-colored scanning electron microscopy (SEM) images of RAW 264.7 cells 1 h after the addition of MSN (red) to the culture media. (**b**) Macrophage with a cluster of MSN (red) on the cell surface. (**c**) Cell filopodia with a bound MSN (pseudo-colored red) at the distal end. (**d**) MSN (red) on cell bodies and TNTs. (**e**) MSN (red) uptake by filopodia projecting from non-adherent cellular bridges (a.k.a. TNTs). * *p* < 0.05; ** *p* < 0.01.

**Figure 3 cancers-12-02892-f003:**
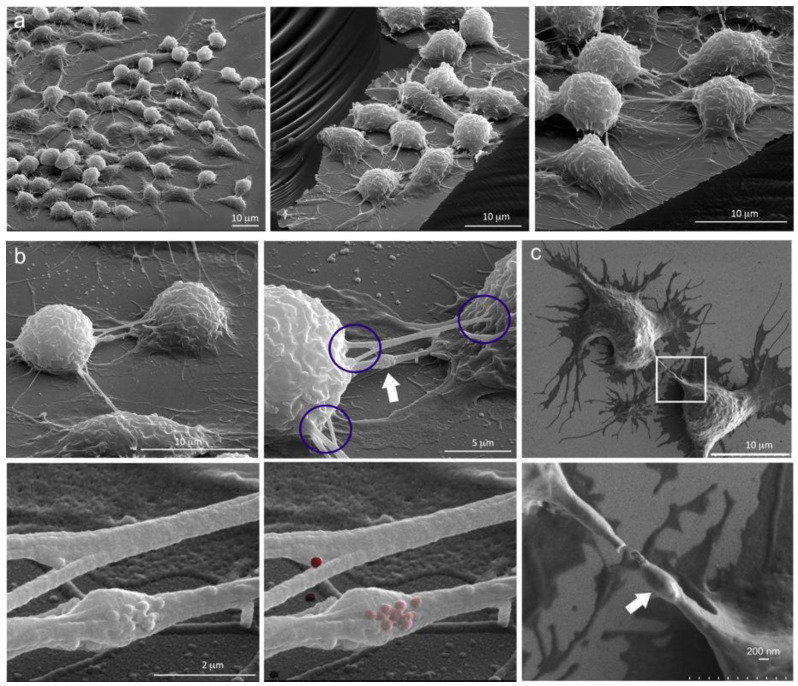
Macrophage TNTs and gondolas. (**a**) Three unique SEM images showing cellular connections among RAW 264.4 cells at increasing magnifications. (**b**) Series of four SEM images showing RAW cells at increasing magnification emphasizing disparate sites of connectivity (circled) and the presence of a gondola within a TNT (white arrow, top right). MSN on the surface of a TNT is shown in grayscale or pseudo-colored red in the lower images (1 h post addition). (**c**) Two macrophages connected by a TNT (low and high magnification). A gondola is highlighted using in the magnified region (white arrow).

**Figure 4 cancers-12-02892-f004:**
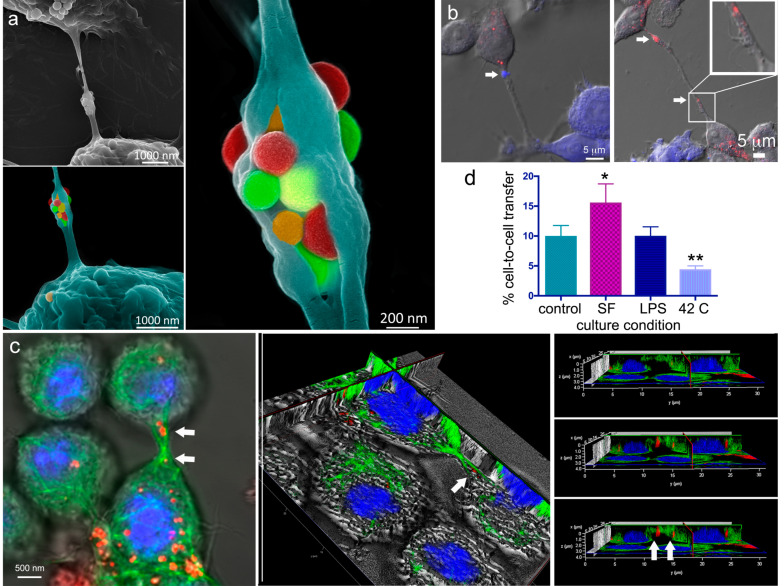
Homotypic MSN transfer between RAW 264.7 macrophages. (**a**) Grayscale and pseudo-colored scanning electron micrographs at three levels of magnification showing false-colored MSN present in a split gondola connecting two macrophages. (**b**) Merged differential interference contrast (DIC) and confocal images of macrophages co-cultured for 24 h after one population was labeled with CellTracker Blue (blue) and the other with DyLight 594 MSN (red). (**c**) Confocal, sectioned, and 3D orthogonal images of macrophages, 5 h following the addition of MSN (red) to the cell media [DIC; Alexa Fluor 488 α-tubulin; DyLight 594 MSN; and nuclei (DAPI, blue)]. The α-tubulin (green) formed cytoskeletal bridges between cells that MSN (red; arrows) utilized to traverse from one cell to another. (**d**) Flow cytometry analysis of relative DyLight 594 MSN transfer from preloaded macrophages to CellTracker Violet positive macrophages under control (NTC), serum-free (SF), lipopolysaccharide (LPS), or hyperthermic (42 °C) conditions. (* *p* < 0.05; ** *p* < 0.01).

**Figure 5 cancers-12-02892-f005:**
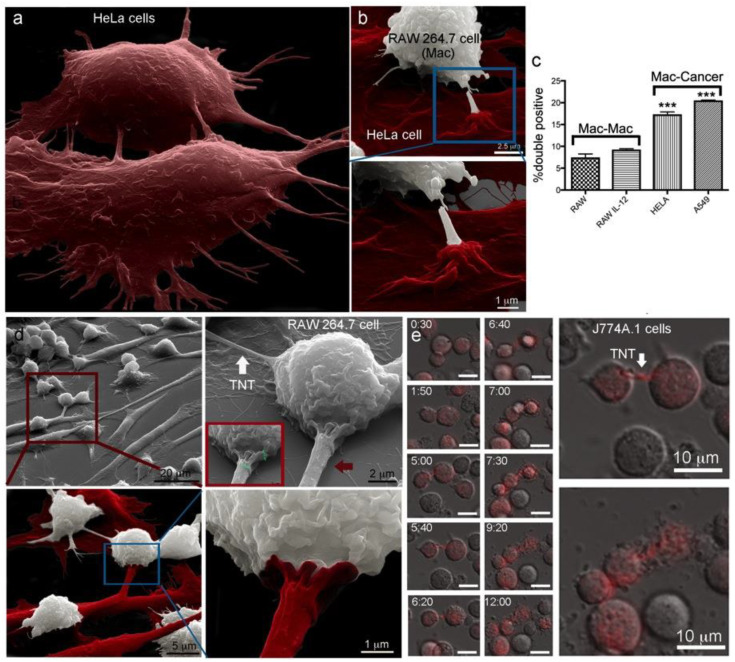
Heterotypic MSN transfer between murine macrophages and human HeLa cancer cells. (**a**) The pseudo-colored scanning electron microscopy (SEM) image shows HeLa cells (red) communicating via tunneling nanotubes (TNTs). (**b**–**d**) SEM and flow cytometry were used to visualize heterotypic TNTs and compare rates of material transfer from RAW 264.7 macrophages to other macrophages or HeLa cells. (**b**) HeLa cells (red) formed elaborate connections with murine macrophages (white) following 24 h co-culture. (**c**) Flow cytometry analysis of relative intercellular transfer of fluorescent MSN among cells initially with single color MSN and the impact of stimulation with IL−12 on transfer among macrophages. (**d**) SEM images showing homotypic connections between macrophages (white arrow) or heterotypic connections between macrophages and cancer cells (red arrow). Inset displays a size marker (green). (**e**) Merged DIC and fluorescent images showing the homotypic transfer of doxorubicin between J774A.1 macrophages at select time points following initiation of co-culture (bar 10 µm in all images) of naïve macrophages with macrophages pretreated with doxorubicin-loaded mesoporous silicon particles. Two of the images are enlarged to the right to highlight the doxorubicin filled TNT (top; arrow) and subsequent cell death (bottom)., *** *p* < 0.001.

**Figure 6 cancers-12-02892-f006:**
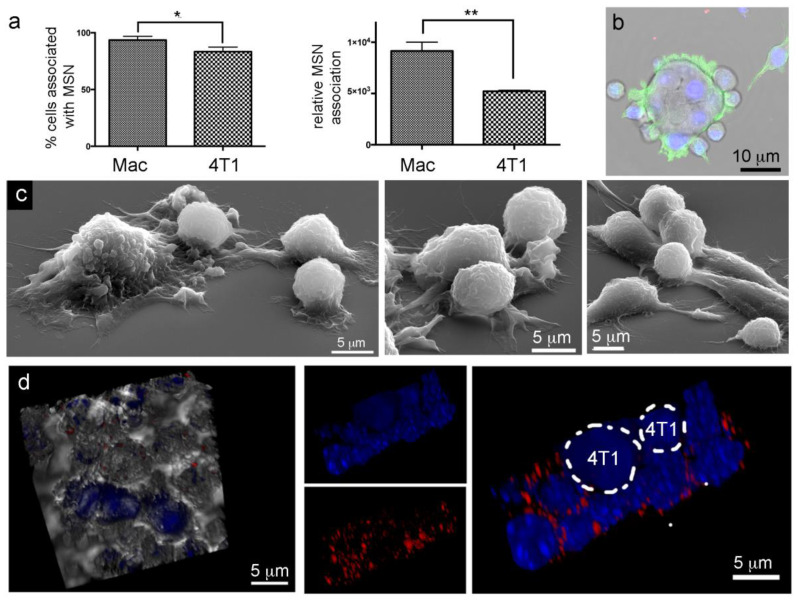
MSN internalization without heterotypic nanoparticle transfer between syngeneic RAW 264.7 macrophages (Mac) and 4T1 cancer cells. (**a**) Flow cytometry analysis of fluorescent MSN uptake presented as percent positive cells and mean uptake per population. (**b**) Merged DIC and fluorescent image of a 4T1 cancer cell surrounded by adherent macrophages (green, phalloidin). (**c**) SEM images of co-cultured 4T1 cancer cells and RAW 264.7 macrophages (round cells) showing direct cellular contact. (**d**) Lack of heterotypic MSN transfer shown using 3D merged DIC and fluorescent images of 24 h co-cultured CellTracker Violet positive 4T1 cancer cells and MSN-loaded (red) macrophages (nuclei, blue, DAPI). * *p* < 0.05, ** *p* < 0.01.

**Figure 7 cancers-12-02892-f007:**
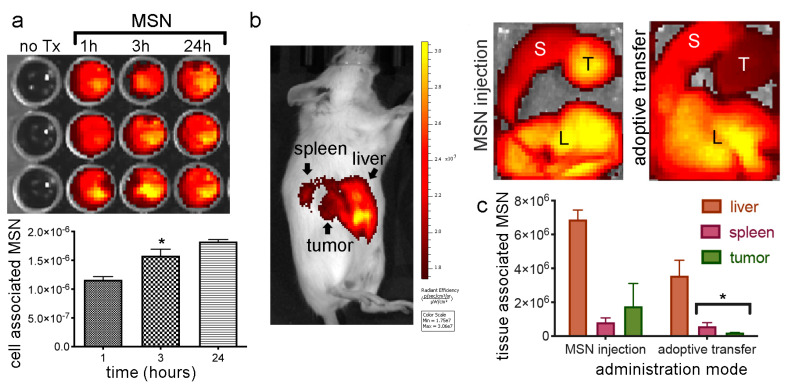
In vivo MSN biodistribution following direct injection or adoptive transfer via splenocytes. (**a**) IVIS Spectrum acquired images and analysis of DyLight 800 MSN uptake by BALB/c splenocytes at 1, 3, or 24 h (fluorescent images and graphed means with standard error bars). (**b**) BALB/c mice injected with DyLight 633 or 800 MSN (nanoparticles; NPs) were imaged at 24 h post-injection using the IVIS Spectrum (L, liver; S, spleen; T, tumor). Images of excised organs from mice injected with free MSN or MSN-loaded splenocytes (cells). (**c**) Average fluorescent intensity of organs (*n* = 3 mice/group) presented graphically 24 h after injection of free MSN or MSN-loaded splenocytes. * *p* < 0.05.

**Figure 8 cancers-12-02892-f008:**
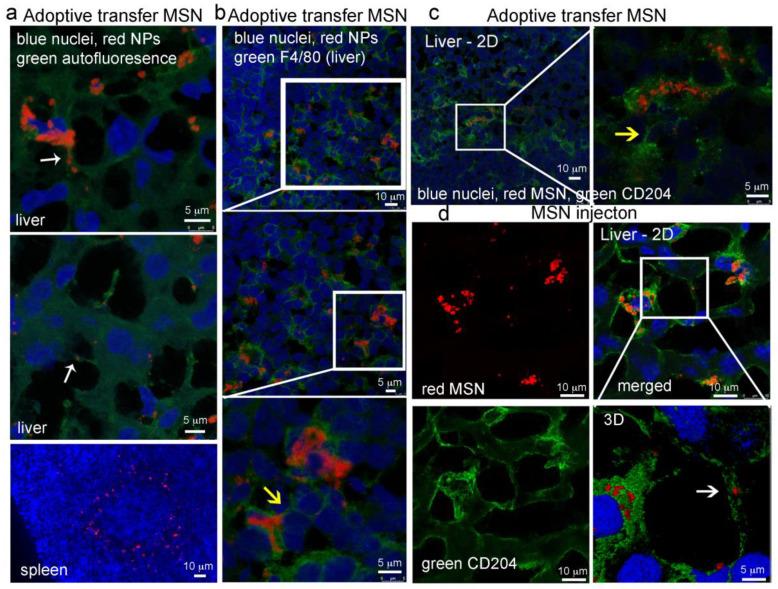
Association of MSN with macrophages in immune-competent syngeneic mice following intravenous injection. BALB/C mice with 4T1 breast tumors were injected intravenously with 250 µg DyLight 633-labeled MSN (red) directly (d, free) or via the adoptive transfer of splenocytes preloaded with MSN (**a**–**c**). Twenty-four-hour post-administration, filtering organs and tumors were excised and labeled with either F4/80 or CD204 fluorescent antibodies (green). (**a**) Liver and spleen with no antibody staining (green, autofluorescence). (**b**) F4/80-labeled (green) tissues. (**c**,**d**) CD204-labeled (green) tissues. Arrows highlight cell-to-cell connections with (white) or without (yellow) MSN.

**Figure 9 cancers-12-02892-f009:**
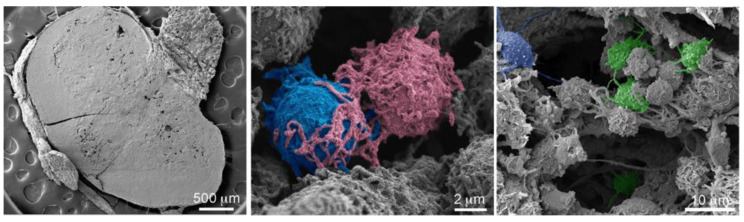
In vivo cellular interactions. Scanning electron micrographs show a sectioned inguinal lymph node (**left**) and higher magnification images of cells within the lymph node excised from a BALB/c mouse with a 4T1 breast tumor. Several cells are pseudo-colored to distinguish the cells and highlight extensive cellular connections between cells in close proximity (**middle**) or at a distance (**right**).

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
