# Peer review of "Direct Transfer of Mesoporous Silica Nanoparticles between Macrophages and Cancer Cells"

_cancers, 2020, doi:10.3390/cancers12102892_

Round 1

Reviewer 1 Report

In this work, the authors evaluated the transfer of mesoporous silica nanoparticles (MSN)  from macrophages to neighboring cells through tunneling nanotubes (TNT)-like structures. I believe that their work is very interesting, clear and well written.

More specifically,

1) they evaluated the internalization of MSN by macrophages (RAW cell line) by FACS analysis and  scanning electron microscopy (Figure 2 and 3). I believe that the images of SEM are beautiful.  I wonder whether the MSNs move along the TNTs outside or inside the membrane. Could the authors address this issue?

2) they observed the MSN transfer between RAW cells by confocal microscopy and SEM, in different conditions.

3) they observed the Heterotypic MSN transfer macrophages and human HeLa cells. In figure 5e, the authors could insert an Annexin V assay of MSN-doxorubicin loaded macrophages.

4) they evaluated the biodistribution of free MSN or MSN-splenocytes and finally they found  adoptive transferred MSN or free MSN associated with F4/80 or CD204 macrophages in liver but no MSN has been detected in tumor.

Author Response

Dear Editors and Reviewers,

Thank your for taking the time to review our manuscript. We greatly appreciate your insightful comments and have edited the manuscript to address your suggestions. The entire manuscript has been edited to correct for grammar/English errors and text has been added to clarify ambiguous sections as described below.

Reviewer #1

In this work, the authors evaluated the transfer of mesoporous silica nanoparticles (MSN) from macrophages to neighboring cells through tunneling nanotubes (TNT)-like structures. I believe that their work is very interesting, clear and well written.

More specifically,

1) They evaluated the internalization of MSN by macrophages (RAW cell line) by FACS analysis and scanning electron microscopy (Figure 2 and 3). I believe that the images of SEM are beautiful.  I wonder whether the MSNs move along the TNTs outside or inside the membrane. Could the authors address this issue?

This is an excellent question. The following text has been added to the manuscript:

Line 191: “The use of SEM surface topography to visualize MSN associated with cells is on occasion ambiguous since MSN seen on the cell surface could be: 1) sitting on cell membrane, 2) in the process of internalization, or 3) exist just beneath the cell membrane (i.e. the dehydration processing causes cell shrinkage which could result in MSN protrusion of cell membranes). Alternative imaging with confocal microscopy supports the presence of MSN both within and on the cell surface. With respect to mobility of MSN, the potential exists for movement both within and on the surface of TNTs. Bacteria have been reported to “surf” on the surface of TNTs based on constitutive flow of the nanotube surface (PMID: 17142745), providing a possible mechanism for external MSN movement. On the other hand, intracellular MSN are associated with microtubules and based on their location in endosomes (PMID: 22891864), transport is likely driven by microtubule motors such as kinesins and dynein, or by actin-based myosin motors (PMID: 28231489). In summary, MSN can traffic both on the surface and within TNTs.”

2) they observed the MSN transfer between RAW cells by confocal microscopy and SEM, in different conditions.

3) they observed the Heterotypic MSN transfer macrophages and human HeLa cells. In figure 5e, the authors could insert an Annexin V assay of MSN-doxorubicin loaded macrophages.

To address the highly significant question of macrophage cell viability in the presence of DOX-loaded MSN, we have added the following text and citation:

Line 328: “Zhang et al. reported that silica-based nanocapsules loaded with doxorubicin can be loaded into macrophages with minimal drug release in the first hours after uptake and minimal effect on in vivo cell migration to tumors (PMID 30368972).”

4) they evaluated the biodistribution of free MSN or MSN-splenocytes and finally they found  adoptive transferred MSN or free MSN associated with F4/80 or CD204 macrophages in liver but no MSN has been detected in tumor.

Reviewer 2 Report

In this paper, Franco et al studied the interesting phenomenon of MSN transfer between macrophages and cancer cells. Although content transfer between cells, especially between immune cells and cancer cells has been described before, it is important to document how different cargos influence and perhaps dictate the transfer process. Overall, the experimental design was good, the data was attractive, and the presentation was informative and organized. There are only a few minor issues with the manuscript.

  1. The font sizes in some of the figures are not too small. It is important to make sure that the texts are legible. For instance, Figure 7b.
  2. A related issue as above, the font sizes in the figures are highly heterogeneous. Some figures have huge fonts and some very small. It's better to keep everything consistent.
  3. It is nice to see pseudo colors, but the designation of different cellular parts seemed not very convincing. For instance, Figure 5b.
  4. It would be better (although not necessary) to provide EDX mapping of the SEM images, so that the MSNs are more visible. 
  5. There are quite a few language errors and ambiguities throughout the paper. For example, line 259-260, did the author want to say that the existence was previously unknown, or that the reason for the existence was unknown? A careful proofreading is necessary.  

Author Response

Dear Editors and Reviewers,

Thank your for taking the time to review our manuscript. We greatly appreciate your insightful comments and have edited the manuscript to address your suggestions. The entire manuscript has been edited to correct for grammar/English errors and text has been added to clarify ambiguous sections as described below.

Reviewer #2

In this paper, Franco et al studied the interesting phenomenon of MSN transfer between macrophages and cancer cells. Although content transfer between cells, especially between immune cells and cancer cells has been described before, it is important to document how different cargos influence and perhaps dictate the transfer process. Overall, the experimental design was good, the data was attractive, and the presentation was informative and organized. There are only a few minor issues with the manuscript.

  1. The font sizes in some of the figures are not too small. It is important to make sure that the texts are legible. For instance, Figure 7b.

The figures were all sized to 6 inches in width and the same font type and size were used to create section letters and headings. Graph axes were resized to similar font size as well. Figure 7 was rearranged to create a figure with a 6-inch width to eliminate unused space in the final version.

  1. A related issue as above, the font sizes in the figures are highly heterogeneous. Some figures have huge fonts and some very small. It's better to keep everything consistent.

Font type and sizes have been adjusted in figures to increase consistency (see #1 for additional details).

  1. It is nice to see pseudo colors, but the designation of different cellular parts seemed not very convincing. For instance, Figure 5b.

I agree that false-coloring is subjective and gray-scale images have been included as well as colored images. In addition, the following statements have been added to the text:

Line: 295: “False-coloring of images is subjective and discrimination of cell type was based on findings that macrophages are relatively small and round compared to cancer cells. In addition, intercellular connections differ between cell types as described in the next paragraph. Gray-scale images are included in addition to false-colored images for mixed culture images.”

  1. It would be better (although not necessary) to provide EDX mapping of the SEM images, so that the MSNs are more visible. 

This is an excellent suggestion and was used with ease to confirm the presence of iron oxide nanoparticles beneath the cell membrane in unpublished data. Unfortunately, we did not perform EDX of the cells treated with MSN when acquiring SEM images.

  1. There are quite a few language errors and ambiguities throughout the paper. For example, line 259-260, did the author want to say that the existence was previously unknown, or that the reason for the existence was unknown? A careful proofreading is necessary.  

My apologies, the “existence” in line 259 was indeed an incorrect word choice. This section of the paragraph has been revised as follows and several authors have reviewed the entire manuscript to correct for English/grammatical errors. Changes are tracked in the manuscript.

Line 319: “Similar to the cells presented in Fig. 5b, Fig. 5d shows an elaborate point of connection between the human cancer cell and the murine macrophage. The elaborate membrane structure exists on the thick adhernet connection and has therefore been colored as originating from the cancer cell. The biological significance of these extravagant connections involved in heterotypic TNT formation beteween macrophages and cancer cells is unknown.”